# The Flow of the Redox Energy in Quercetin during Its Antioxidant Activity in Water

**DOI:** 10.3390/ijms21176015

**Published:** 2020-08-21

**Authors:** Zhengwen Li, Mohamed Moalin, Ming Zhang, Lily Vervoort, Erik Hursel, Alex Mommers, Guido R. M. M. Haenen

**Affiliations:** 1Department of Pharmacology and Toxicology, Faculty of Health, Medicine and Life Sciences, Maastricht University, P.O. Box 616, 6200 MD Maastricht, The Netherlands; maxamed.moalin@gmail.com (M.M.); z.ming@maastrichtuniversity.nl (M.Z.); l.vervoort@maastrichtuniversity.nl (L.V.); hursel.erik@gmail.com (E.H.); a.mommers@maastrichtuniversity.nl (A.M.); 2Research Centre Material Sciences, Zuyd University of Applied Sciences, 6419 DJ Heerlen, The Netherlands

**Keywords:** flavonoid, quercetin, redox modulation, antioxidant, molecular mechanism, electron transfer, proton transfer

## Abstract

Most studies on the antioxidant activity of flavonoids like Quercetin (Q) do not consider that it comprises a series of sequential reactions. Therefore, the present study examines how the redox energy flows through the molecule during Q’s antioxidant activity, by combining experimental data with quantum calculations. It appears that several main pathways are possible. Pivotal are subsequently: deprotonation of the 7-OH group; intramolecular hydrogen transfer from the 3-OH group to the 4-Oxygen atom; electron transfer leading to two conformers of the Q radical; deprotonation of the OH groups in the B-ring, leading to three different deprotonated Q radicals; and finally electron transfer of each deprotonated Q radical to form the corresponding quercetin quinones. The quinone in which the carbonyl groups are the most separated has the lowest energy content, and is the most abundant quinone. The pathways are also intertwined. The calculations show that Q can pick up redox energy at various sites of the molecule which explains Q’s ability to scavenge all sorts of reactive oxidizing species. In the described pathways, Q picked up, e.g., two hydroxyl radicals, which can be processed and softened by forming quercetin quinone.

## 1. Introduction

In all life forms, opposing forces provide the energy that flows through networks in an organism, which fuels life [1]. The major source of energy in our body is provided by the redox reactions of glycolysis, lipid oxidation, and the citric acid cycle [2]. This redox energy flows through biochemical networks and is the driving force of vital cellular reactions. However, a substantial part of the energy flow is uncontrolled and produces reactive oxidizing species that can inflict damage to crucial cellular components such as lipids, proteins and DNA. This plays a central role in numerous patho-physiological processes, from ageing to inflammation [1,3].

Flavonoid antioxidants can neutralize these reactive oxidizing species by donating electrons. By accepting electrons, reactive oxidizing species are converted into relatively unreactive compounds, that do not damage the cell. For example, by accepting an electron of a flavonoid, the hydroxyl radical is converted into the hydroxyl anion that, after protonation, becomes water. Their redox modulating activity has been implicated in the beneficial health effect of flavonoid antioxidants [4,5,6].

Although the antioxidant activity of flavonoids has been studied for over several decades, the molecular mechanism remains enigmatic. One of the reasons for this is that in addressing their antioxidant activity, all flavonoids are generally put under the same umbrella. Nevertheless, it is known that a minor change in the molecular structure of a flavonoid can drastically change its antioxidant activity [7], indicating that the molecular mechanism of each flavonoid is unique. A “universal” mechanism for all flavonoids does not exist. Another reason for the enigmatic nature of the antioxidant activity is that the antioxidant effect of a specific flavonoid comprises of a series of sequential reactions, and each reaction in this series has its specific characteristics [8]. Not separating these reactions will blur the vision on the molecular mechanism.

In the present study we aim to zoom in on the antioxidant activity of one of the most important flavonoids, i.e., quercetin (Q). By combining experimental data with quantum calculations, it is possible to look at the sequential reactions in its antioxidant activity separately. In this way we aim to unveil how the redox energy flows though the molecule during its antioxidant activity.

Most of the studies on the antioxidant mechanism of quercetin only address one electron oxidation, from Q to the Q radical (Q^•^) [9,10]. The oxidation of Q is, however, a two-electron process, in which Q is finally converted into a quinone methide (QQ) [11]. Our previous work also confirmed the formation of QQ in the radical scavenging reactions of Q, although the evidence still is indirect [12,13]. So ultimately, the redox energy picked up by Q flows through the molecule to form QQ. Via QQ the redox energy is processed further in the antioxidant network. In the present manuscript we first examine Q itself, and the molecular mechanism of the scavenging of radicals by Q. The site at which the electron is donated is also determined. Subsequently, we examine Q^•^, where its unpaired electron is retained within the molecule. We also address the molecular mechanism of the scavenging of radicals by Q^•^ and at which site Q^•^ donates an electron. Finally, we focus on the two-electron oxidation product, i.e., quercetin quinone methide (QQ), how QQ is formed, and how the energy is stabilized in this molecule. Sharpening the picture will help us to understand the molecular mode of action of polyphenols and to see differences between polyphenols, so we will be able to select the appropriate redox modulating compound for a specific redox-mediated disorder.

Three theoretical pathways exist for the molecular mechanism of an antioxidant such as Q. These are hydrogen atom transfer (HAT), single electron transfer followed by proton transfer (SET-PT), and sequential proton loss electron transfer (SPLET). The descriptors characterizing the enthalpy of the first step of each of these three mechanisms are the homolytic bond dissociation enthalpy (BDE) for HAT, the ionization potential (IP) for SET-PT, and the proton affinity (PA) for SPLET [14,15,16,17]. These reaction enthalpies were calculated as the difference of the calculated enthalpies of the reactant (which is fully protonated Q) with that of the products. These products are the Q radical (Q^•^) and a hydrogen atom (H^•^) for BDE; the protonated Q radical (Q^•+^), and an electron (e) for IP, and deprotonated Q (Q^−^) and a proton (H^+^) for PA. The equations are:BDE = *H*(Q^•^) + *H*(H^•^) − *H*(Q)(1)
IP = *H*(Q^•+^) + *H*(e) − *H*(Q) (2)
PA = *H*(Q^−)^ + *H*(H^+^) − *H*(Q) (3)

In these equations, *H* represents the enthalpies of the corresponding fragments.

We intend to study the antioxidant activity of Q in the water phase, and Q is partly deprotonated at physiological pH. Because deprotonation, and the group that deprotonates, play pivotal roles in the antioxidant activity of Q, we also addressed this in our study. We will take the deprotonated conformation of Q into account. Q has five phenolic OHs, which have five different pKa values [18]. The pKa of Q has been studied using various methods, but no consistent value was found; namely, it varies from 6.74 [19] when determined by spectrophotometry, to 7.19 [20] when determined by capillary zone electrophoresis, and 7.71 [21]/8.21 [22] when determined by potentiometry. There is also no consensus on the pKa of the second acid group of Q, the reported value of this pKa varies from 8.4 to 9.4 [18]. Moreover, there is also no consensus on which group of Q is the most acidic one.

DFT (density functional theory) calculation by Markovic et al. [23], at the M05-2X/6-311+G(d,p) level, showed that the proton affinity of the 5, 7, 3′, 3, and 4′ hydoxyl groups of Q was 26.75, 22.45, 27.71, 25.80, and 22.21 Kcal/mol, respectively, suggesting that the 4′-OH is deprotonated first, followed by the 7-OH. Calculation at lower basis set of 6-31+G(d,p) by Vásquez-Espinal et al. [15], revealed that the proton affinity of the 5, 7, 3′, 3, and 4′ hydoxyl groups of Q were 28.30, 23.50, 28.59, 26.39, and 23.20 Kcal/mol, respectively. This calculation also suggests that the 4′-OH group is the most acidic one. However, some studies, especially experimental studies, conclude that the 7-OH group is the most acidic group [24,25]. The most convincing is the ^13^C CP/MAS NMR (Cross Polarization/Magic Angle Spinning Nuclear Magnetic Resonance) study performed by Musialik et al. [25]. They found that the ^13^C-7 signal of Q in D_2_O was the first one that considerably shifted toward higher frequency field by raising the pH, indicating that the 7-OH group is the most acidic one. To solve the inconsistency in the literature, we decided to perform time-dependent density functional theory (TDDFT) calculations to obtain the UV spectrum of 7-OH deprotonated Q (7DE) and the UV spectrum of 4′-OH deprotonated Q (4′DE). Subsequently, we compared the obtained calculated spectra with experimental UV spectra of a Q solution in water recorded at various pH values.

Recently, a quantum chemical computational study by Brovarets et al. [26], on the tautomerization of Q by intramolecular proton transfer, revealed 48 stable conformers (24 planar and 24 non-planar) with relative Gibbs free energies within the range of 0.0 to 25.3 Kcal/mol. This included conformers formed by proton transfer from the 3-OH group to the 4 oxygen atom with a Gibbs free energy change of around 13 Kcal/mol. This proton transfer will affect the antioxidant mechanism of Q. Additionally, proton transfer from the 5-OH group to the 4 oxygen atom is relevant. Therefore, these two proton transfers will be taken into consideration in our study.

In the literature on the antioxidant activity of Q, the importance of C2-C3 double bond of Q is frequently emphasized. Therefore, the present study will also address this, in order to sharpen our view on how the redox energy flows through Q during its antioxidant activity.

## 2. Results

### 2.1. The Molecular Mechanism of Scavenging of the First Radical by Q—Deprotonation of Q

To determine which mechanism is involved in the scavenging of the first radical by Q, the O-H bond dissociation enthalpy (BDE), ionization potential (IP), and proton affinity (PA) of Q were calculated to estimate the likelihood of HAT, SET-PT and SPLET, respectively. These reaction enthalpies were calculated as the difference of the calculated enthalpies of the reactant (which was Q) with that of the products. All possible products formed out of Q by the three mechanisms are depicted in Figure 1, and the enthalpies of the formation of these products out of Q are shown. Generally, the lower the value of the enthalpy, the more likely it is that the products will be formed.

The IP value of Q to Q^•^ is 112.51 Kcal/mol, which is the highest of all three types of calculated enthalpies. This indicates that SET-PT is not an energy favorable pathway for the first radical scavenging.

The calculated BDE values of the OH groups show that HAT from the 4′-OH group, with an enthalpy of 77.49 Kcal/mol, is preferred over HAT from the other OH groups. Nevertheless, the BDE of the 4′-OH group is substantial higher than that of the PA values, indicating that HAT is also not favorable from an energy perspective.

The sequence of the calculated PA values of the hydroxyl groups from low to high are: 26.11, 26.80, 31.20, 31.27 and 31.39 Kcal/mol, for the 4′-OH, 7-OH, 3-OH, 5-OH and 3′-OH groups, respectively.

It can be concluded that based on the enthalpies of three descriptors, the deprotonation of Q, represented by the PA values, is preferred in an aqueous solution. This indicates that SPLET is the preferred mechanism and that the first step in the scavenging activity of Q in an aqueous solution is deprotonation of Q.

### 2.2. From Q to Q Anion—Deprotonation of the 7-OH Group

The PA values of the 7-OH and 4′-OH groups are the lowest. The difference of both PA values is only 0.69 Kcal/mol, so it is hard to determine which hydroxyl group is deprotonated first. As outlined in the introduction, in the literature there is also no consensus on the order of deprotonation of the hydroxyl groups of Q. To solve this, the UV/Vis spectrum of 7-OH deprotonated Q (7DE) and that of 4′-OH deprotonated Q (4′DE) were calculated. Since tautomerization of 7-OH deprotonated Q has been proposed, this was also considered. The tautomer in which the proton of the 5-OH group is transferred to the oxygen atom at the 4-position appeared to be unstable. Therefore, only the UV/Vis spectrum of the tautomer in which the proton of the 3-OH group is transferred to the 4 oxygen atom (abbreviated as 7DE-ITPT, in which ITPT stands for Intramolecular Proton Transfer) was calculated. These calculated spectra were compared to change in the spectrum of a solution of Q in water when the pH increases (Figure 2).

In acidic solutions with a pH below 5, the spectrum of Q is independent of the pH value. This indicates that at these acidic pH values, practically all Q is fully protonated. The absorption spectrum of the fully protonated Q only has two peaks, which are at 367 nm and 255 nm. Raising the pH above 5 results in the deprotonation of Q as the absorption spectrum changes. There is a red shift of the Q peak of 367 nm of about 10 nm when the pH changes from 5 to 8. Additionally, a new absorption peak appears at around 318 nm (Figure 2).

The simulation results depicted in Figure 2 show that fully protonated Q has a peak in the simulated absorption spectrum at 367 nm, which is practically similar with the wavelength of the peak at 377 nm of the experimental absorption spectrum of fully protonated Q. The calculated spectrum of 7DE has a peak at 376 nm, which is 9 nm higher than the peak of 367 nm of the calculated spectrum of fully protonated Q. This fits with the red shift of 10 nm of the peak of the Q solution at 367 nm caused by raising the pH. Moreover, the new peak that appears in the Q solution at around 318 nm by increasing the pH nicely coincides with the peak in the calculated spectrum of 7DE at that wavelength. In addition, the lower calculated height of the peak around 376 nm in the calculated spectrum of 7DE compared to the height of the peak around 367 nm in the calculated spectrum of Q, is in accordance with the difference in height of the corresponding peaks in the spectrum of the Q solution observed by raising the pH. 4DE has two peaks at 270 nm and 400 nm, which does not fit with the in the change in the spectrum of the Q solution caused by increasing the pH. These observations indicate that the 7-OH group is the most acidic group of Q. Therefore, we consider that deprotonation of the 7-OH group is the first step in the antioxidant activity of Q in water.

### 2.3. From 7-OH Deprotonated Q to Q Radical—Electron Transfer

According to the SPLET mechanism, the second step in radical scavenging of Q is the transfer of an electron from 7DE to an oxidizing species. The enthalpy of the electron donation was calculated. The tautomer of 7DE in which the proton of the 3-OH group is transferred to the 4 oxygen atom (7DE-ITPT) was also taken into consideration.

The enthalpy of the electron transfer of 7DE to the corresponding Q radical (denoted as 7DE-ET, in which ET stands for Electron Transfer) is 107.22 Kcal/mol (ionization potential 2, abbreviated as IP2). Tautomerization of 7DE and subsequent electron transfer (IP3) of this tautomer (giving the 7DE-ITPT-ET) has a total enthalpy of 107.91 Kcal/mol, which is almost the same as the IP2 pathway. This indicates that both pathways are possible. Although the enthalpy gap of 7DE-ET with the 7DE-ITPT-ET is only 0.36 Kcal/mol, their 3D structure greatly differs (Appendix A). The dihedral angle of C2-C3-C1′-C2′ (which reflects the angle of the B-ring with the AC-ring) of the 7DE-ET is 27.65°, while that of the 7DE-ITPT-ET is only 3.7° which indicates that 7DE-ET and 7DE-ITPT-ET do not convert in each other. This means that the pathway is split at 7DE, as 7DE and 7-hDE-ITPT continue to produce different conformers of the Q radical that do not interconvert. Both pathways will be further investigated in the following sections.

In the Q molecules, the electrons in the π orbitals of all the oxygen atoms are part of a conjugated π-system. Therefore, a “deprotonated OH group” (incorrectly) suggesting that the negative charge (electrons) is exclusively on the π-orbital of the oxygen atom, is actually equal to an “oxo group” with the negative charge delocalized in the π-system. “Deprotonated OH group” is used in the text when in the reaction discussed, the OH group referred to, losses its proton. Because we choose to show the carbonyls in the chemical structure of the molecule, and the limitation of the traditional way molecules are depicted, the number and the fixed position of the double bounds in the rings is not accurate. These “errors” are seen in some of the structures shown in Figure 3, Figure 5 and Figure 6. This is the reason we show their 3D structures in the Appendix A.

### 2.4. Electron Structure of the Two Conformers of the Q Radical—Importance of C2-C3 Bond

To determine where the electron was lost from the deprotonated forms of Q, electron density difference maps of 7DE as well as 7DE-ITPT with their corresponding radical were generated.

The electron density difference of 7DE with the 7DE-ET radical shows that by electron transfer, most of the electron density is lost on C8, C10 and O7, and on the C2-C3 bond (Figure 4). This indicates that the electron was primarily donated at these positions. The electron density difference of 7DE-ITPT with 7DE-ITPT-ET shows that in 7DE-ITPT, the electron was primarily donated at C2-C3 bond and O3.

The spin density map shows that, after donating one electron, the unpaired electron in 7DE-ET is mostly delocalized on the AC-ring and concentrated on C8, C10 and O7, and the C2-C3 bond. The relatively high dihedral angle of 7DE-ET of 27.65°, as mentioned in Section 2.3, is in line with the observed delocalization of its unpaired electron primarily in the AC-ring. Although in 7DE-ITPT-ET, the unpaired electron is concentrated on the C-ring and B-ring (especially on the C2-C3 bond), the unpaired electron is delocalized over the whole molecule. The angle in 7DE-ITPT-ET between the AC-ring and B-ring of only 3.7° indicates that the molecule is virtually planar, which is in line with the observed delocalization of its unpaired electron over the π system of the whole molecule.

### 2.5. From Q Radical to Q Radical Anion—Deprotonation of the 3-OH or 4′-OH Group

The 7DE-ET radical and 7DE-ITPT-ET radical, formed when Q has scavenged a radical, can scavenge a second oxidizing species. Of the BDE, IP, and PA values of the radicals (Appendix A), the PA values are significantly lower than the other two, indicating that the molecular mechanism of the antioxidant activity of the Q radicals is also a SPLET mechanism. This means that here also the first step is deprotonation. To determine which hydroxyl group is deprotonated first, the PA values of the hydroxyl groups of both conformers of the Q radical were calculated.

As shown in Figure 5, the enthalpy needed for deprotonation of the 3-OH or 4′-OH group of the 7DE-ET are the lowest and do not differ much, indicating that one of these groups is the first to be deprotonated. The same goes for the enthalpy needed for deprotonation of the 4-OH or 4′-OH group of 7DE-ITPT-ET.

Therefore, the conclusion can be made that for 7DE-ET, as well as 7DE-IPET-ET, the most favorable pathways from an energy perspective are deprotonation of either their OH group in the C-ring or their 4′-OH group in the B-ring. The products in which the 4′-OH group is deprotonated are denoted by 7DE-ET-4′DE and 7DE-IPET-ET-4′DE. It is noteworthy that the products in which the OH group in the B-ring is deprotonated, denoted by 7DE-ET-3DE and 7DE-IPET-ET-4DE in Figure 5, are identical. For the sake of clarity, 7DE-ET-3DE is used in this manuscript to denote this conformer in the next step of the antioxidant activity (the second electron transfer), but it should be realized that this conformer can be formed by two pathways. Therefore, in total, three forms of the Q radical anion need to be considered.

The spin density maps show that the unpaired electron is mostly concentrated on the B-ring and the C2-C3 bond of the Q radical anions. In 7DE-ET-4′DE and the 7DE-ITPT-ET-4′DE, the B-ring is favored. This is probably because in these radical anions, an OH group in the B-ring (i.e., the 4′-OH) is deprotonated. In the 7DE-ET-3DE radical, the C2-C3 bond is favored. This is probably due to the oxo group on the C2-C3 bond.

### 2.6. From the Q Radical Anion to the Q Quinone—Electron Transfer

The next step is that the Q radical anion donates an electron to the second oxidizing species. In this reaction a Q quinone/quinone methide (QQ) is formed.

It was found that the enthalpy difference between each Q radical anion and its corresponding quinone are comparable and have a value of approximately 100 Kcal/mol (Figure 6). 

The electron density difference maps of the Q radical anions and their corresponding quinones appear to be very similar to the spin density map of the Q radical anions (Figure 7).

The electron density difference maps show that the electron is primarily donated at the B-ring and the C2-C3 bond of the Q radical anions. This coincides with the locations at which the spin density in the Q radical anion is the highest.

The total enthalpies of the three pathways from Q leading to QQ are 259.56 Kcal/mol, 252.53 Kcal/mol, and 250.58 Kcal/mol, for 7DE-ET-4′DE-ET, 7DE-ET-3DE-ET, and 7DE-ITPT-ET-4′DE-ET quinone, respectively. That these total enthalpy values do not differ much indicates that all three pathways should be considered for the flow of the redox energy of Q during its antioxidant activity. 

The HOMO-LUMO gap (Appendix A) of the three quinones are 3.16 ev (7DE-ET-4′DE-ET), 3.77 ev (7DE-ET-3DE-ET) and 4.13 ev (7DE-ITPT-ET-4′DE-ET), respectively. These results indicate that the quinones are reactive and that 7DE-ET-4′DE-ET and -ET-3DE-ET quinone are less stable than 7DE-ITPT-ET-4′DE-ET Quinone. Previously, we tried to detect the formation of QQ during the oxidation of Q using HPLC analysis with UV detection, but this failed. This confirms the limited stability of QQ, probably because QQ reacts quickly with water. The indirect evidence for the formation of QQ is that QQ can be trapped with GSH. 6GS-Q and 8GS-Q are exclusively formed [11]. That only these adducts are formed, is consistent with the formation of 7DE-ET-ITPT-4′DE-ET as the most prominent form of QQ. Using mass spectrometry, we were able to detect QQ during the oxidation of Q (Appendix A). To our knowledge, this is the first time that QQ was directly detected in the oxidation of Q.

It is worth mentioning that the dihedral angle of C2-C3-C1′-C2′ of 7DE-ET-ITPT-4′DE is 2° (Appendix A), indicating that this quinone is virtually flat and that the electrons in the π system of the quinone can delocalize over the whole molecule. This indicates the ‘soft’ nature of QQ. That QQ prefers to react with the “soft” GSH over the “hard” ascorbate [27], confirms the soft nature of QQ.

## 3. Discussion

In its antioxidant activity, Q easily picks up redox energy, which is softened when it flows through the molecule, and finally the redox energy is passed over from the oxidized Q into the antioxidant network. The aim of our study is to find, from an energy perspective, the most favorable pathway for the flow of the redox energy through Q during this activity. Usually, only the one electron reaction of Q to form Q^•^ is examined. However, Q donates two electrons in its antioxidant activity [12]. Therefore, it is also examined how Q^•^ can pick up redox energy, and how the energy flows through Q^•^ to form QQ.

In most theoretical studies on the antioxidant activity of Q, the likelihood of HAT, SET-PT and SPLET as the first step is investigated [28,29]. It is worth mentioning that, although the HAT, SET-PT, and SPLET pathways are the most commonly discussed pathways, other pathways have also been proposed [30], such as proton coupled electron transfer (PCET) [31,32]. In PCET, an electron and a proton are transferred in a single kinetic step, but the electron and the proton can come from a different orbital or space. In HAT, one of the PCET pathways, the electron and proton are joined together from the donor to the acceptor. Other pathways, such as sequential proton loss hydrogen atom transfer (SPLHAT), are also considered [33]. The features of these pathways have been nicely described by Galano and Alvarez-Idaboy [34]. They pointed out that in the PCET pathways the antioxidant activity largely depends on the electronegativity of the H donor and acceptor, while in the SPLET pathways, the characteristics of the solvent are pivotal in the first step of the antioxidant reaction. The first step of different mechanism has been done by calculating the enthalpy or Gibbs free energy of the first step in the three mechanisms in various solvents using different calculation methods at numerous levels. A consensus seems to be that the enthalpy for the first step in SPLET—that is, PA—is the lowest [9]. We also found that the enthalpy for PA is the lowest, indicating that the deprotonation of Q is most energy favorable first step. This is in line with the observation that Q in water at pH 2.1, when it is fully protonated, is a poor antioxidant compared to Q in water at pH 7.4, when Q is partially deprotonated [35]. This confirms that deprotonation is the primary first step in the antioxidant activity of Q in the water.

There is no consensus on which OH group of Q is the most acidic. Our calculation is in agreement with other DFT calculations that the 4′-OH will be the first group of Q to deprotonate. However, the difference between the PA of the 4′-OH group and the 7-OH group is small, and experimental studies indicate that the 7-OH group is the most acidic group [25]. Of our TDDFT simulated UV spectra, the difference between the spectrum of Q and 7DE fitted the best with the change in the spectrum of a Q solution caused by raising the pH. This confirms the conclusion of the previously reported experimental studies that the 7-OH group is the most acidic.

A shortcoming of the DFT calculation we use is that a hydrogen bond of the OH groups of Q with water cannot be reflected or simulated by the implicit SMD (Solvation Model based on Density) model we used, or other implicit solvent models since no real water molecules had been added. The 4′-OH group can form an intramolecular hydrogen bond with the 3′-OH group, which may lead to the enhancement of their stability. The deprotonated 7-OH group cannot form an intramolecular hydrogen bridge. This means that the formation of an intermolecular hydrogen bridge with a water is, from the energy perspective, more relevant for 7DE than for 4′DE. This may explain why based on the calculation, the 4′-OH seems to be the most acidic, while actually—as found in the experimental studies—the 7-OH is the most acidic. Therefore, we considered that deprotonation of the 7-OH group is the first step in the antioxidant activity of Q. The DFT calculation shows that after deprotonation, a tautomer of 7DE can be formed by in which the proton of the 3-OH group is transferred to the 4 oxygen atom (7DE-ITPT) [26]. Therefore, the formation of this tautomer is also considered.

In the SPLET mechanism, the next step after the deprotonation is electron donation. Our electron density difference maps reveal that 7DE as well as its tautomer 7DE-ITPT can donate an electron at various sites of the molecule with variable energies, which explains Q’s ability to scavenge all sorts of reactive oxidizing species.

The dihedral angle of C2-C3-C1′-C2′ (the angle between the B-ring and AC-ring) greatly affects the delocalization of the unpaired electron. In 7DE-ET, the dihedral angle is 27.6°, which prevents the delocalization of the unpaired electron from passing through the C2-C1′ bond and concentrates the unpaired electron on B-ring of 7DE-ET. The angle in 7DE-ITPE-ET is 3.7° (Appendix A), which indicates that it is almost planar, and that the unpaired electron can delocalize over the whole molecule. The relatively high difference in the dihedral angle also means that the conformers of the Q radical formed out of 7DE and 7-DE-ITPT do not interconvert. Therefore, the pathway splits after the deprotonation of Q, and 7DE and 7-DE-ITPT continue to produce a different conformer of the Q radical.

The spin density map of both Q radicals shows that the spin density of the unpaired electron concentrates near the deprotonated OH and oxo groups, i.e., on C2-C3 bond and O5 for 7DE-ITPT-ET and C8 and O7 for 7DE-ET. The concentration of the unpaired electron near deprotonated OH groups and oxo groups is also found in the Electron Spin Resonance spectra of Q and Q derivatives recorded at high pH [36]. This can be explained by the electron-withdrawing nature of the deprotonated OH group and the oxo group [37].

It is known that the oxidation of Q is a two-electron process, which finally generates QQ [38]. Nevertheless, the antioxidant activity of Q^•^ has not been studied well. The enthalpy of PA of the hydroxyl groups of both Q radicals (Appendix A), are significantly lower than that of BDE or IP, indicating that molecular mechanism of the antioxidant activity of Q^•^ is also SPLET.

Our calculations show that the enthalpy for deprotonation of the OH group in the C-ring or the 4′-OH group in the B-ring of both Q radicals are the lowest and do not differ much. Interestingly, deprotonation of the OH group in the C-ring of both radicals, giving the products denoted as 7DE-ET-3DE in one pathway and as 7DE-ITPT-ET-4DE in the other pathway, appears to generate the same molecule. This indicates that both pathways can come together again. Deprotonation of the 4′-OH generates different molecules. Therefore, in addition to the mutual pathway, there are two other pathways (Figure 8). The angle in 7DE-ITPE-ET, 7DE-ET-3DE, and 7DE-ITPT-ET-4’DE is 3.7°, 0.03°, and 0.001° (Appendix A), which indicates that these molecules are almost planar and explains that the unpaired electron can delocalize over the whole molecule, which will reduce the reactivity.

The spin density map shows that in 7DE-ET-4′DE and 7DE-ITPT-ET-4′DE, the deprotonated 4′-OH group concentrates the unpaired electron on B-ring. Additionally, this can be explained by the electron-withdrawing nature of this group [36]. Notably, in all radicals of Q, the unpaired electron is for a substantial part concentrated on the C2-C3 bond. Moreover, the sp2 nature of the C2-C3 bond is essential for connecting the π-system of the AC-ring with that of the B-ring. In addition, the electron density difference maps show that the first electron as well as the second electron is donated for a substantial part from the C2-C3 bond. This all may explain why the importance of the C2-C3 moiety is highlighted in virtually all the reports on the antioxidant activity of Q [38].

Although the enthalpy difference between the three quinones formed is relatively small, their abundance in the solution might greatly differ and 7DE-ITPT-ET-4′DE-ET is the most abundant one. This is in agreement with previous work [27].

It has to be realized that Q has the capacity to react with up to 12 radicals in antioxidant capacity assays [39]. This indicates that QQ and oxidation products of QQ also have antioxidant activity. The molecular mechanism of their antioxidant activity is beyond the scope of the present study. Moreover, QQ quickly reacts with GSH and ascorbate in biological systems before QQ is oxidized [11,12]. Therefore, the oxidation of QQ and its oxidation products is considered to be less relevant.

## 4. Computation Details 

The geometry and vibration spectra of Q and oxidized forms of Q, as well their deprotonated forms, were generated by Gaussian 09 [40] at M06-2X [41]/6-311+G(d,p) [42] level. Calculation of the enthalpy of the various forms of Q in gas phase were carried out at M06-2X/def2-TZVP [43] level with a D3 correction [44] by Orca [45]. The solvent effect was included by using the implemented SMD (water) method [46]. Since the solvent model cannot exactly calculate the enthalpy of solvation (H(solvation)), the solvation free energy was employed as an approximate alternative. For this, the recommended basis set M05-2X/6-31G* was used. Additionally, 1.89 Kcal/mol was added to balance the difference of the enthalpy of the molecules in gas phase with that in the water phase. The ZPE (zero-point energy) correction factor is 0.97, as described by Truhlar [47]. The enthalpy of an electron, hydrogen atom or proton in the gas phase, used in the present manuscript are 0.751 Kcal/mol, −313.75 Kcal/mol and 1.48 Kcal/mol, respectively [48]. The solvation enthalpy of an electron, hydrogen atom or proton are −18.51 Kcal/mol, −0.955 Kcal/mol and −252.365 Kcal/mol, respectively [49]. All enthalpies and free energies were calculated for 298.15 °K. The spin density maps and electron density difference maps were generated by Multiwfn [50] and VMD [51]. The 3D structures of all investigated compounds can be seen in Appendix A. Here in the IP is the adiabatic IP.

Time-dependent density functional theory (TDDFT) was applied to simulate the UV/VIS spectrum of Q and deprotonated Q. Herein, functional PEB0 at the Def2-tzvp level was used. To obtain accurate spectra over a wide range of wavelengths, in total 10 excited states were taken into consideration [52]. The spectra were generated by Multiwfn by the Gaussian function with the full width at half maximum (FWHM) of 0.67 ev [35]. The spectra of a 50 μM Q in a 50 mM phosphate buffer at the indicated pH were recorded on a Cary 50 spectrophotometer. At a pH above 7.5, 50 μM of ascorbate was added to prevent or reverse the oxidation of Q. Due to the absorption of ascorbate below 300 nm, only the absorption spectrum above 300 nm is given when ascorbate was present.

## 5. Conclusions

The antioxidant activity of flavonoids has been studied for a long time. At first it was suggested that for all flavonoids proceeded by the same simple scavenging reaction:Flavonoid-OH + R^•^ → Flavonoid-O^•^ + RH

This evolved when the antioxidant activity of more flavonoids was tested, and descriptive Structure Activity Relationships were constructed, which also formed a theoretical fundament. Now it is known that the molecular mechanism of the antioxidant activity of each flavonoid is unique, and the simple scavenging reaction is split up into sequential reactions when the redox energy flows through the molecule.

By combining previously reported and new experimental data with quantum calculations, the present study confirms that SPLET is the mechanism of the antioxidant activity of Q, as well as that of Q^•^. The enthalpy of the second SPLET process of Q^•^ (115.87 Kcal/mol) is less than that of the first ‘SPLET’ process of Q (134.71 Kcal/mol), suggesting that the second radical scavenging reaction of Q is quicker than the first scavenging, which is in line with the literature. This means that in studying the antioxidant activity of Q, two sequential SPLET mechanisms should be considered. Mostly, the antioxidant activity of Q^•^ is not considered, indicating that only half of the antioxidant activity of Q is studied.

Moreover, the series of sequential reactions in the antioxidant activity of Q are separately studied and connected. The calculations show that Q can pick up redox energy at various sites of the molecule, with variable energies. This explains Q’s potent scavenging ability of all sorts of reactive oxidizing species. Moreover, it appears that not one, but several pathways are possible for the flow of redox energy through Q. In addition, these pathways are also intertwined. This flexibility of the flow of the redox energy is depicted in the overview Figure (Figure 8). There were already some clues that pointed towards the flexibility of the antioxidant activity of Q: Q contains more than one antioxidant “pharmacophore”, and these pharmacophores do not work separately, but interact [8,53]. The flexibility of the flow of the redox energy through the molecule adds to the efficiency and increases the versatility of Q’s redox modulating potency.

Using mass spectrometry, we were able to detect QQ during the oxidation of Q. Moreover, our study confirms the importance of the C2-C3 moiety in the antioxidant activity of Q.

## Figures and Tables

**Figure 1 ijms-21-06015-f001:**
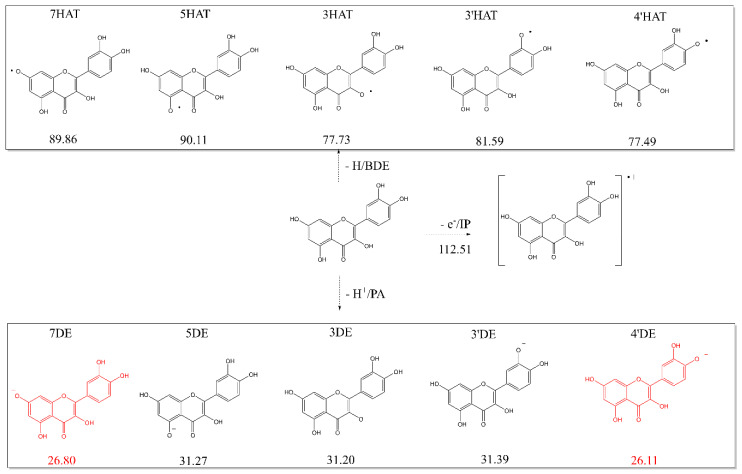
The products formed out of Q by hydrogen atom transfer, electron transfer or proton transfer; and the BDE, IP or PA enthalpies (in Kcal/mol) for the formation of these products. The two products in red are the products formed out of Q by deprotonation (proton transfer) of its 7-OH or 4′-OH group, and the formation of these products appeared to require the lowest reaction enthalpy.

**Figure 2 ijms-21-06015-f002:**
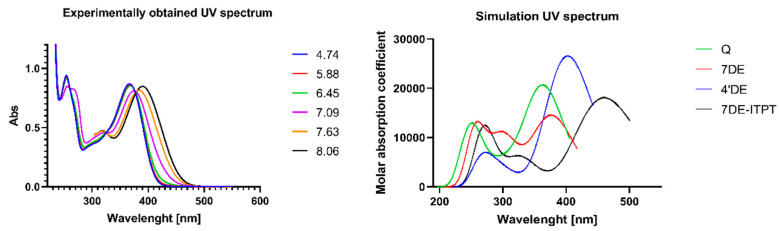
The experimentally obtained UV spectrum of a Q solution (50 μM) in a 50 mM phosphate buffer at various pH values (**left** panel) and the simulated UV spectra (**right** panel) of Q and the three energetically most favorable mono-deprotonated forms of Q. i.e 7-OH deprotonated Q (7DE), 4′-OH deprotonated Q (4′DE) and the tautomer of 7DE in which the proton of the 3-OH group is transferred to the 4 oxygen atom (7DE-ITPT).

**Figure 3 ijms-21-06015-f003:**
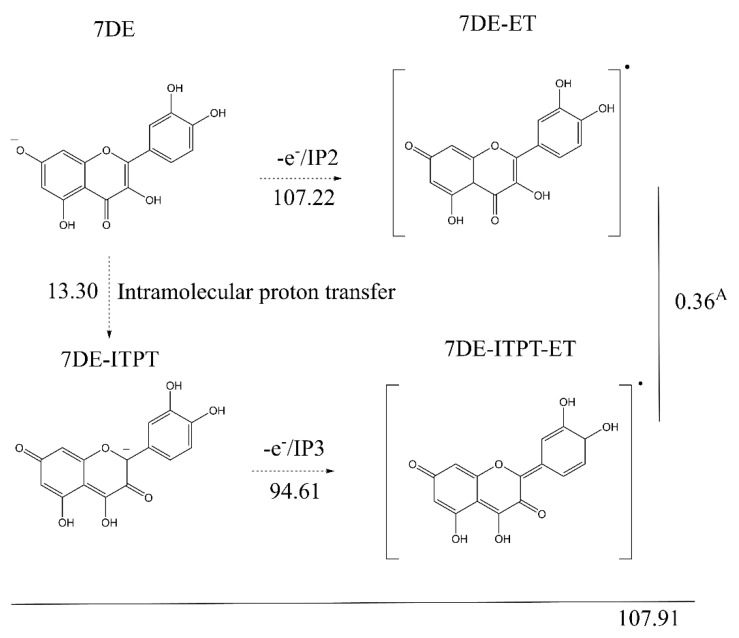
The two pathways for the electron transfer of 7-OH deprotonated Q (7DE) to a Q radical. The enthalpies requirement (Kcal/mol) of the reactions are given. The first pathway is a direct electron transfer from 7DE giving 7DE-ET. The second pathway is first an intramolecular proton transfer from 7DE to give 7DE-ITPT, followed by an electron transfer from 7DE-ITPT to give 7DE-ITPT-ET. The enthalpy difference of 0.36 Kcal/mol between the two conformers of the Q radicals, 7-DE-ET and 7DE-ITPT-ET, is also depicted.

**Figure 4 ijms-21-06015-f004:**
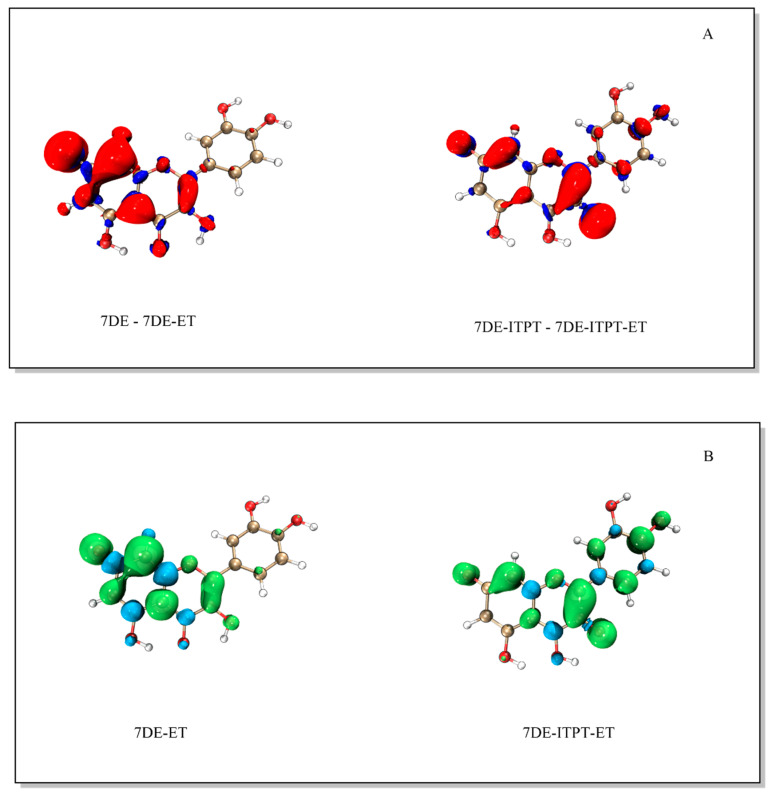
The electron density difference map of 7DE and of 7DE-ITPT with their corresponding radical (Panel (**A**)), and the spin density of the 7DE-ET and the 7DE-ITPT-ET radical (Panel (**B**)). In constructing the electron density difference map, the radical formed immediately after the electron transfer has taken place is used, and the electrons in the radical have not yet relaxed. The difference map shows where the electron is lost. The electron density map of the radicals depicted in panel B is of the stable radical.

**Figure 5 ijms-21-06015-f005:**
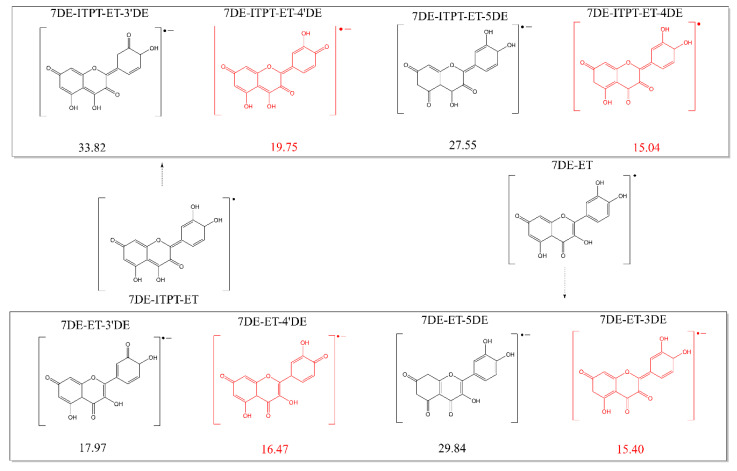
The enthalpy requirement of the deprotonation of the two conformers of the Q radical, 7DE-ET and 7DE-ITPT-ET. The four products in red are the products formed out of both Q radical conformers by deprotonation (proton transfer) of their 3-OH or 4′-OH group, and the formation of these products appeared to require the lowest reaction enthalpy.

**Figure 6 ijms-21-06015-f006:**
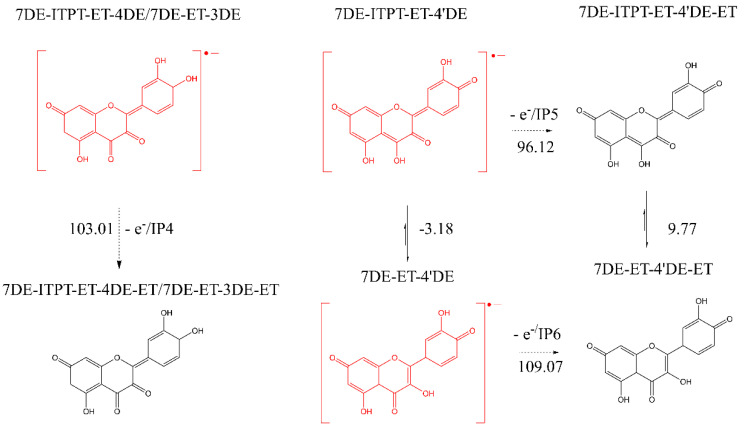
The enthalpy requirement (in Kcal/mol) of the second electron donation from the three conformers of the Q radical to form the corresponding quinone.

**Figure 7 ijms-21-06015-f007:**
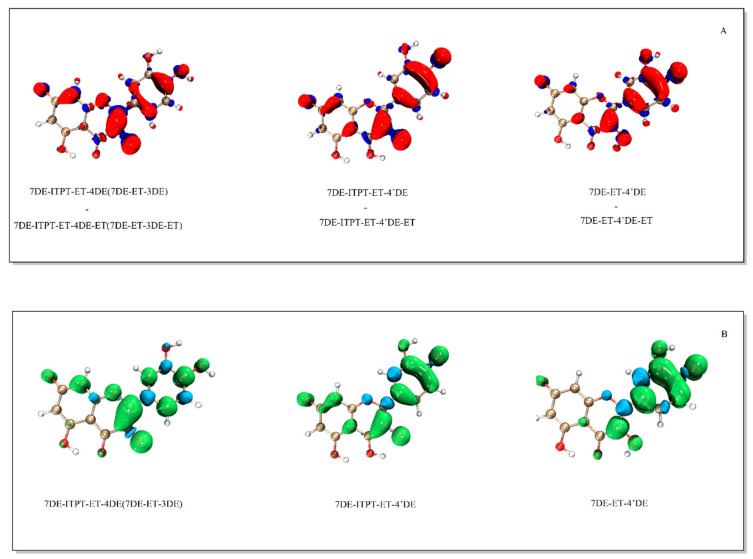
The electron density difference of the three forms of the Q radical with their corresponding quinone (panel (**A**)) and the spin density of the three forms of the Q radical (panel (**B**)).

**Figure 8 ijms-21-06015-f008:**
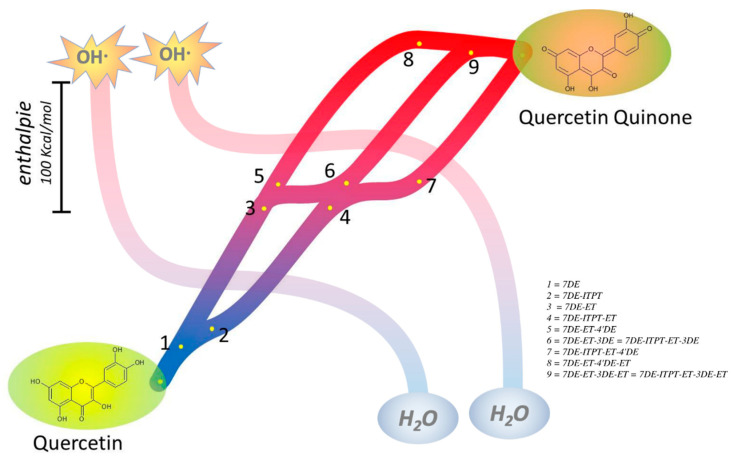
Overview of the main pathways of the flow of the redox energy in Q during its antioxidant activity in water. First, Q is deprotonated at the 7 position to give 7DE (**1**). Then the pathway spits, due to an intramolecular proton transfer, generating 7DE-ITPT (**2**). In one pathway, electron transfer coverts 7DE in the Q radical 7DE-ET (**3**). In 7DE-ET the second deprotonation can take place at the 4′ or 3 position, again splitting the pathway, generating 7DE-ET-4′DE (**5**) and 7DE-ET-3DE (**6**), respectively. After the second electron transfer, the quinones 7DE-ET-4′DE-ET (**8**) and 7DE-ET-3DE-ET (**9**) are formed. The other pathway starts with the generation of 7DE-ITPT (**2**). Electron transfer converts 7DE-ITPT in the Q radical 7DE-ITPT-ET (**4**). The second deprotonation of this Q radical (**4**) also can take place at two positions. Interestingly, deprotonation of the Q radical (**4**) at the 4 position generates a molecule identical to 7DE-ET-3DE (**6**), and here the pathways come together. The second deprotonation of the Q radical (**4**) at the 4′ position generates a third pathway by the formation of 7DE-ITPT-ET-4′DE (**9**). After a second electron transfer 7DE-ITPT-ET-4′DE is converted in the quinone 7DE-ITPT-ET-4′DE-ET (Quercetin Quinone). Of the quinones formed, 7DE-ITPT-ET-4′DE-ET (Quercetin Quinone) has the lowest energy content, and ultimately this appears to be the most abundant quinone in the antioxidant activity of Q. In these pathways, Q can pick up the hard energy of e.g., two hydroxyl radicals, and the hard energy is softened before it is passed on into the antioxidant network.

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
