# Peer review of "The Flow of the Redox Energy in Quercetin during Its Antioxidant Activity in Water"

_ijms, 2020, doi:10.3390/ijms21176015_

Round 1
Reviewer 1 Report
The manuscript “The flow of the redox energy in Quercetin during its antioxidant activity in water” presents a computational study of the different sequential reactions that are implied in the antioxidant activity of quercetin. Computational data are compared with previously published experimental results of UV spectrometry, and mass spectrometry is used to confirm the formation of a quinone methide from quercetin, which supports the proposed sequential reactions of the authors for explaining the antioxidant activity of quercetin. The topic may be interesting for the readers, however manuscript suffers some flaws that need to be remedied.
1. The abstract should specify that the work presented is mainly a computational study and that experimental data used to contrast computational results, come from previous works. I also suggest to include in the abstract that sequential routes proposed for explaining the antioxidant activity of quercetin were supported by an experimental identification of a quinone methide from quercetin, using mass spectrometry, which is the new experimental result that generates this work.
2. The authors describe three possible theoretical pathways to explain the molecular mechanism of antioxidants. Although in the literature these three pathways are the most commonly discussed, other pathways have been proposed too, e.g. Proton coupled electron transfer (PCET), which although gives the same products that HAT mechanism, is possible of distinguishing by computational calculations. Considering that this work aboard the issue mainly from a theoretical view of point, I suggest improving the discussion of mechanism, incorporating these elements. The following reference can help to authors about this issue: A.Galano, J.R. Alvarez-Idaboy Int J Quantum Chem. 2019; 119: e25665.
3. The computational methodology needs to be clarified. The authors do not specify in the methodology section which kind of IP is calculated. Two approaches are used to obtain this parameter, adiabatic or vertical. This should be explicit in the manuscript, considering that if exist significant differences for geometries of the radical species regards to the neutrals, the vertical IP is not appropriate, and the adiabatic approach must be considered. If the authors calculated VIP, they should obtain AIP for a couple of structures and see if they approach is justify or, if not, authors must calculate adiabatic IPs. I suggest considering the same reference given above.
Some minor remarks:
Line 334-336. The authors mention that a shortcoming of the DFT calculation is that hydrogen bonds among OH groups of Q with water cannot be reflected or simulated. However, it is possible to include explicit solvent molecules, regardless of the use of the implicit solvent model. Do not include explicit solvent molecules is a decision of the authors (which depend on the computational costs, or other variables that they can consider). The authors need to amend this paragraph to avoid this confusion.
Line 95. “M0-52X” must be changed to M05-2X (keep same notation in all the manuscript, also for M06-2X)
Line 126. “andthe enthalpies” must be changed to “and the enthalpies”
Author Response
Reply to reviewer 1.
First of all, we would like to thank the reviewer for the time spent in reviewing our manuscript, and for the constructive comments.
The comments of the reviewer and our response:
General comment. The manuscript “The flow of the redox energy in Quercetin during its antioxidant activity in water” presents a computational study of the different sequential reactions that are implied in the antioxidant activity of quercetin. Computational data are compared with previously published experimental results of UV spectrometry, and mass spectrometry is used to confirm the formation of a quinone methide from quercetin, which supports the proposed sequential reactions of the authors for explaining the antioxidant activity of quercetin. The topic may be interesting for the readers, however manuscript suffers some flaws that need to be remedied.
Reply: We acknowledge the view of the reviewer and will use more the detailed comments of the reviewer (see the next items) to improve our manuscript.
Comment 1. The abstract should specify that the work presented is mainly a computational study and that experimental data used to contrast computational results, come from previous works. I also suggest to include in the abstract that sequential routes proposed for explaining the antioxidant activity of quercetin were supported by an experimental identification of a quinone methide from quercetin, using mass spectrometry, which is the new experimental result that generates this work.
Reply: The reviewer correctly mentions that the abstract does not give the points given in the comment. Please note that we also present new experimental results (the spectra of Q recorded at several pH values, and the mass spectrometry results on the formation of QQ). Please also note that according to the guidelines of the Journal, the abstract may only have 200 words. We struggled to make an abstract that meets this requirement (it contains 192 words) that best covers the main message of our manuscripts. By adding the points suggested by the reviewer, we have to remove information of the abstract that (in our opinion) is more essential than the points raised by the reviewer. We hope that after this explanation, the reviewer can agree with our abstract.
The points raised by the reviewer are addressed in the conclusion paragraph of our manuscript. In the revised manuscript, we have revised our conclusion paragraph in such a way that the points (correctly) mentioned by the reviewer are more clearly addressed in the conclusion paragraph.
Revised conclusion:
By combining previously reported and new experimental data with quantum calculations, the present study confirms that SPLET is the mechanism of the antioxidant activity of Q as well as that of Q•……..(page 13,line 411)
Moreover, the series of sequential reactions in the antioxidant activity of Q are separately studied and connected. ….(page 13,line 449)
Using mass spectrometry, we were able to detect QQ during the oxidation of Q. Moreover, our study confirms the importance of the C2-C3 moiety in the antioxidant activity of Q.(page 13,line 459-460)
Comment 2. The authors describe three possible theoretical pathways to explain the molecular mechanism of antioxidants. Although in the literature these three pathways are the most commonly discussed, other pathways have been proposed too, e.g. Proton coupled electron transfer (PCET), which although gives the same products that HAT mechanism, is possible of distinguishing by computational calculations. Considering that this work aboard the issue mainly from a theoretical view of point, I suggest improving the discussion of mechanism, incorporating these elements. The following reference can help to authors about this issue: A.Galano, J.R. Alvarez-Idaboy Int J Quantum Chem. 2019; 119: e25665.
Reply: The reviewer mentioned the pathways beyond the three most commonly discussed and suggested improving the discussion of mechanism which is helpful and for the benefit of the readers we added the following paragraph in the Discussion section.
(Page 11 Line 320-330)
It is worth to mention that, although the HAT, SET-PT, and SPLET pathways are the most commonly discussed pathways, also other pathways have been proposed [30], such as proton coupled electron transfer (PCET)[31, 32]. In PCET an electron and a proton are transferred in a single kinetic step, but the electron and the proton can come from a different orbital or space. In HAT, one of the PCET pathways, the electron and proton go joined together from the donor to the acceptor. Other pathways, such as sequential proton loss hydrogen atom transfer (SPLHAT,) are also considered [33]. The features of these pathways have been nicely described by Galano and Alvarez‐Idaboy [34]. They pointed out that in the PCET pathways the antioxidant activity largely depends on the electronegativity of the H donor and acceptor, while in the SPLET pathways, the characteristics of the solvent are pivotal in the first step of the antioxidant reaction.
Comment 3. The computational methodology needs to be clarified. The authors do not specify in the methodology section which kind of IP is calculated. Two approaches are used to obtain this parameter, adiabatic or vertical. This should be explicit in the manuscript, considering that if exist significant differences for geometries of the radical species regards to the neutrals, the vertical IP is not appropriate, and the adiabatic approach must be considered. If the authors calculated VIP, they should obtain AIP for a couple of structures and see if they approach is justify or, if not, authors must calculate adiabatic IPs. I suggest considering the same reference given above.
Reply: We do account geometry relaxation in all our calculations and this is why we use enthalpy but not electron energy to describe all the reaction. As kindly mentioned by reviewer we do forget to stress this, and which could cause a misunderstanding.
As suggested by the reviewer, we have specified that we used the adiabetic approach to obtain the IP (page 13, Line 421)
Some minor remarks:
- Line 334-336. The authors mention that a shortcoming of the DFT calculation is that hydrogen bonds among OH groups of Q with water cannot be reflected or simulated. However, it is possible to include explicit solvent molecules, regardless of the use of the implicit solvent model. Do not include explicit solvent molecules is a decision of the authors (which depend on the computational costs, or other variables that they can consider). The authors need to amend this paragraph to avoid this confusion.
- Line 95. “M0-52X” must be changed to M05-2X (keep same notation in all the manuscript, also for M06-2X)
- Line 126. “andthe enthalpies” must be changed to “and the enthalpies”
Reply:
- As correctly mentioned by the reviewer, we used a solvent model to incorporate the effect of the solvent. We indeed decided not to include solvent molecules, because – as also mentioned by the reviewer – this will increase the computational costs. Moreover, including solvent molecules will increase the complexity of the model, and in our opinion, this is (not yet) validated for us. In our revised manuscript we will mention that we used the implicit solvent model to avoid to delivery wrong message to the readers. (page 11 ,line 345 and 346.)
- We have corrected the errors in line 95 (ant the rest of the manuscript) and line 126 (now 127).

Reviewer 2 Report
The work is a continuation of the research of the team, recently published, which is of great importance because it clarifies what happens during the process of antioxidant activity of flavonoids, and in the present work is the flavonol quercetin in an aqueous medium. It is easy to read and understand the work and concepts. Only very few points (they are well known for those are working with this issues but maybe unknown for other ones.
In line 95 the term DFT calculation appears at the first time. The meaning of the abbreviation is needed. The same for the term CP/MAS NMR. The same for line 335: SMD. The same in line 404: ZPE.
Author Response
Reply to reviewer 2.
First of all, we would like to thank the reviewer for the time spent in reviewing our manuscript, and for the constructive comments.
The comments of the reviewer and our response:
General comment: The work is a continuation of the research of the team, recently published, which is of great importance because it clarifies what happens during the process of antioxidant activity of flavonoids, and in the present work is the flavonol quercetin in an aqueous medium. It is easy to read and understand the work and concepts. Only very few points (they are well known for those are working with this issues but maybe unknown for other ones.
Reply: We would like to thank the reviewer for expressing the stimulating appreciation for our manuscript.
Specific comment: In line 95 the term DFT calculation appears at the first time. The meaning of the abbreviation is needed. The same for the term CP/MAS NMR. The same for line 335: SMD. The same in line 404: ZPE.
Reply: The comments made are appreciated and corrections are made according to the suggestion of the reviewer.

Round 2
Reviewer 1 Report
I recommend the acceptance of the manuscript as it stands